

# Microphysical characteristics of precipitation within convective overshooting over East China observed by GPM DPR and ERA5

Nan Sun[1], Gaopeng Lu[1], Yunfei Fu[1]

[1]School of Earth and Space Sciences, University of Science and Technology of China, Hefei, 230026, China

*Corresponding to*: Yunfei Fu, fyf@ustc.edu.cn

**Abstract.** We examine geographical distribution pattern of convective overshooting and its internal microphysical three-dimensional structure of precipitation over East China by matching Global Precipitation Measurement Dual-frequency Precipitation Radar instrument (GPM DPR) with European Centre for Medium-Range Weather Forecasts 5th Reanalysis (ERA5). Convective overshooting events mainly occur over NC (Northeast China) and northern MEC (Middle and East China), with a magnitude of only $10^{-3}$; Radar reflectivity of convective overshooting over NC accounts for a higher proportion below the zero level, while MEC and SC (South China) account for a higher proportion above the zero level, indicating stronger upward motion and more ice crystal particles; The microphysical processes within convective overshooting are unique, leading to various properties of the droplets in precipitation. Droplets of convective overshooting are large, but sparse. And its effective radius of droplet, below 10 km altitude, is almost exceeding 2.5 mm, which is about twice than normal precipitation. Convective overshooting humidifies air below the cloud top and obviously increases the ozone near tropopause as a result of influx of ozone from lower troposphere and sinking of air with high concentration ozone in the stratosphere. Findings of this study may have important implications for the microphysical evolution associated with convective overshooting, and provide more accurate precipitation microphysical parameters as the input of the model simulation.



## 1 Introduction

Convective overshooting provides a rapid transport mechanism that can irreversibly transport water vapor and chemical constituents from lower troposphere to the upper troposphere and lower stratosphere (UTLS) by mixing them with environmental air (Fueglistaler et al., 2004; Frey et al., 2015), which has a direct impact on radiation balance and global climate change (Solomon et al., 2010). As one of the main sources of ozone destroying OH hydroxyl radicals, stratospheric water vapor can help to destroy ozone, which has potential effects on radiative forcing (Anderson et al., 2012). Previous studies show that convective overshooting has a net dehydrating effect on the stratospheric humidity (Danielsen, 1993; Sherwood and Dessler, 2001). Recently, modeling and observational studies show the moistening effect of convective overshooting on the stratosphere (Chaboureau et al. 2007; Jensen et al. 2007; de Reus et al. 2009; Avery et al. 2017) because of the injection of ice mass into the stratosphere (Grosvenor et al., 2007; Corti et al., 2008; Chemel et al., 2009; Khaykin et al., 2009). In addition to these impacts on water vapor, the convective overshooting affecting on the UTLS temperature has also attracted much attention (Sherwood et al., 2003; Chae et al., 2011; Biondi et al., 2012).

In addition to UTLS composition effects, convective overshooting is often associated with severe and hazardous weather (e.g., heavy rain, hail, tornadoes, and strong winds) at the Earth's surface with important impacts on society and economy (Line et al., 2016; Bedka et al., 2018; Marion et al., 2019). Given these potentially significant impacts, it's of high importance to understand the characteristics of convective overshooting, which attract considerable attention in recent years (Johnston et al., 2018; Muhsin et al., 2018).

Perhaps one of the most poorly understood features of convective overshooting is the microphysical structure of precipitation, such as particle size, concentration, phase state and other parameters. Understanding the microphysical characteristics of convective overshooting is helpful to clarify the efficiency of water vapor transported to the lower stratosphere by convective overshooting. In addition, the microphysical processes within convective overshooting are closely related to storm dynamics and thermodynamics through latent heat, and the quantitative description of microphysical characteristics is helpful to improve the accuracy of model simulation parameters (Homeyer and Kumjian, 2015). Liu et al. (2012) studied the climatological characteristics of convective overshooting and found that rain rates of convective overshooting are bigger than that of deep convection. Homeyer and Kumjian (2015) observed



the radar reflectivity characteristics within the convective overshooting from the analysis of the
polarimetric radar. Although the above studies have explored the characteristics of some precipitation
parameters within convective overshooting, we still lack the understanding of more precipitation
microphysical parameters and more detailed microphysical processes within convective overshooting
due to the limitations of observation methods.
To fully study the microphysical characteristics of convective overshooting, accurate methods of
detecting the frequency and long-term distribution of convective overshooting are required. The
traditional ways for detecting convective overshooting is to find pixels with brightness temperatures
colder than a given temperature threshold (Machado et al. 1998; Rossow and Pearl 2007). Gettelman et al.
(2002) have studied the cloud regions colder than the tropopause temperature on infrared images and
found that the frequency of tropical convective overshooting is about 0.5%. However, it is impossible to
guarantee that the low value of infrared brightness temperature represents clouds penetrating the
tropopause rather than cirrus or cloud anvil in the upper air due to the lack of vertical structure
information of convection. With the launch of Precipitation Radar aboard Tropical Rainfall Measuring
Mission (TRMM), three-dimensional structure information of precipitation within the convective
overshooting can be provided (Alcala and Dessler, 2002; Liu and Zipser, 2005) and a new method for
detecting the convective overshooting is proposed that is to fine pixels with rain top height higher than
tropopause height (Xian and Fu, 2015; Sun et al., 2021), which improve the accuracy of detecting
convective overshooting. Still, TRMM PR can't provide the precipitation microphysical information,
which limits our study on the internal microphysical structure within convective overshooting. Besides,
TRMM PR can underestimate the height of convective overshooting because of only sensitive to large
precipitation particles (sensitivity at ~17 dBZ) (Hanii and Zheng, 2014).
As the continuation of TRMM PR, Global Precipitation Measurement (GPM) carrying the first
Dual-frequency Precipitation Radar (DPR) launched in February 2014. GPM DPR include two bands of
precipitation radar, which provides excellent opportunities for studying the microphysical structure of
precipitation (Sun et al., 2022a). Liu et al., (2016, 2019) have used GPM KuPR and ERA-Interim
6-hourly data set to study climatology and detection of convective overshooting. However, the above
studies only use the KuPR data and mainly focus on the geographical distribution, the microphysical
processes of convective overshooting remain unknown. Besides, the matching time between GPM and
ERA-Interim is too long (6 hour) to ensure the accuracy of convective overshooting detection.



Another difficulty in convective overshooting detection is to obtain tropopause height data with high
spatial and temporal resolution. On the one hand, the determination of the tropopause is still under debate.
At present, the following two definitions of the tropopause are widely adopted throughout the world: one
is the cold tropopause, and the other is the thermodynamic tropopause. However, the cold point
tropopause is only physically meaningful in the latitude zone 10 °S-10 °N near the equator (Highwood
and Hoskins, 1998; Rodriguez-Franco and Cuevas, 2013). Therefore, this paper uses the thermodynamic
tropopause, which is defined by the World Meteorological Organization (WMO) (WMO, 1957). The
thermodynamic tropopause is based on the temperature lapse rate, also known as lapse-rate tropopause.
The accurate calculation of the tropopause height based on this definition, on the other hand, depends on
the temperature profile data with high spatial and temporal resolution. The latest generation of reanalysis
data ERA5 provides hourly estimates of a large number of atmospheric, land and oceanic climate
variables, which has attracted much attention due to its much higher spatial and temporal resolution than
its predecessor ERA-Interim, especially in the upper troposphere and lower stratosphere (Hoffmann et al.
2019). Sun et al. (2022b) verified the accuracy for the tropopause height calculated from temperature
profiles of ERA5 by comparing ERA5 with other popular datasets.
East China is located in the East Asian monsoon region, with unique climate characteristics. The
precipitation of East China in summer is affected by the circulation anomalies of the East Asian tropical
and subtropical monsoon and their interactions. The precipitation anomalies not only have an important
impact on industrial and agricultural production, social infrastructure construction, but also threaten the
safety of human life and property. Many scholars have studied the characteristics of precipitation in East
China (Zhang et al., 2018; Xu, 2020), but few have studied the characteristics of convective overshooting
and its internal precipitation microphysical structure over East China. The purpose of this study is to
examine the microphysical characteristics of convective overshooting over East China by matching the
precipitation data from GPM DPR and meteorological parameters from ERA5.
**2 Data and method**
**2.1 DPR-based precipitation dataset**
GPM DPR include KuPR (Ku band, 13.6 GHz) and KaPR (Ka band, 35.5 GHz), two bands of
precipitation radar. KuPR is similar to TRMM PR and has a longer wavelength, which is better at



detecting heavy precipitation (the minimum detected precipitation is about 0.5 mm/h). However, KaPR
has a shorter wavelength, which is more sensitive to weak precipitation (the minimum detected
precipitation is about 0.2 mm/h). Based on the different echo characteristics of Ku band and Ka band, the
dual channel inversion algorithm can be used to retrieve DSD (Droplet Size Distribution). Here we use
the precipitation datasets are provided by the GPM level 2 product 2ADPR in version 6 from 2014 to
2020 in summer (June, July and August). The horizontal resolution is 5 km and the vertical resolution is
125m. The precipitation microphysical parameters provided by GPM 2ADPR include droplet
concentration ($dBN_0$) and effective radius ($D_0$).
**2.2 ERA5-based meteorological dataset**
The meteorological data are from ERA5 reanalysis datasets. And the following parameters are used in
this paper: temperature, specific humidity, vertical velocity, ozone mass mixing ratio, U-component of
wind, and V-component of wind. The time resolution is 1 h and the horizontal resolution is 0.25 °×0.25 °.
**2.3 Definition of the convective overshooting**
The convective overshooting is defined as the storm top height above the real-time tropopause height.
storm top height is obtained from the GPM DPR. Tropopause height is calculated from the temperature
profiles from ERA5 according to the definition from the World Meteorological Organization, whose
characteristics are as follow: (1) the atmospheric lapse rate is 2 K km−1 or less and (2) the atmospheric
lapse rate does not exceed 2 K km−1 between the tropopause level and all higher levels within 2 km
(WMO, 1957).
**2.4 Study areas**
The study areas are marked as black boxes in Fig. 1a and only the land parts are studied. To have a better
understanding of precipitation microphysical structure over different regions of East China, we divided
the study areas into three parts according to its climatic characteristics and previous studies (Sun et al.,
2022a). From north to south, they are NC (Northeast China; 38 °–50 °N, 118 °–130 °E), MEC (Middle and
East China; 26.5 °–38 °N, 112 °–123 °E), and SC (South China; 18 °–26.5 °N, 108 °–123 °E)。



## 3 Results

### 3.1 Case studies

Three cases selected from NC, MEC and SC are analyzed to lay a foundation for the subsequent statistical analysis. The precipitation characteristics of the three cases are shown as Fig. 2. The Case 1 (C1) occurs in NC. A total of 65 pixels in which convective overshooting occurs and their rain rate are mostly over 20 mm/h (Fig. 2a) and their storm top height are over 12 km (Fig. 2b). The strong radar reflectivity along A1B1 occurs at 35-95km away from point A1, and the strongest echo is up to 50 dBZ, appears at 0-5 km (Fig. 2c). The maximum echo height is about 15 km, 2 km higher than the tropopause height. The Case 2 (C2) occurs in MEC. There are 58 pixels in which convective overshooting occurs and their rain rate are more than 25 mm/h (Fig. 2d) and their storm top height are mostly over 14 km (Fig. 2e). The radar echo along A2B2 is very strong and the strongest echo is up to 50 dBZ, which is about 45-95 km away from point A2 (Fig. 2f). The highest echo can reach to about 17 km altitude. The Case 3 (C3) occurs in SC. There are 8 convective overshooting pixels in C3 and their rain rate is over 20 mm/h (Fig. 2g) and their storm top height is between 14 and 18 km (Fig. 2h). The strongest echo occurs at 60-70 km away from point A3 and the highest echo can reach to 17.2 km, about 0.5 km higher than the tropopause height (Fig. 2i).

To learn about the characteristics of large scale circulation of cases, we calculate the distribution of Precipitable Water Vapor (PWV), streamlines and Vertical Velocity (VV), shown as Fig. 3. In general, the area in which convective overshooting occurs have abundant PWV and strong ascending movement. In C1, PWV is between 20 and 65 mm. The PWV of the area where the precipitation case occurs (big black box) is between 40 and 55 mm. The PWV of the region in which convective overshooting occurs is obviously higher than else region, which is between 50 and 55 mm (Fig. 3a). The vertical upward movement near the convective overshooting is strong, range between -0.03 and -0.12 Pa/s (Fig. 3b). In C2, The PWV of the area where the precipitation case occurs is between 50 and 65mm. The PWV of the area in which convective overshooting occurs are between 50 and 55 mm (Fig. 3c). The VV near the convective overshooting is mostly between -0.09 and -0.15 Pa/s (Fig. 3d). In C3, the PWV near the precipitation area and convective overshooting area are both between 65 and 75 mm (Fig. 3e), which are relatively high. The vertical upward movement near the precipitation area and convective overshooting area are very strong and the VV are between -0.12 and -0.18 Pa/s.



**3.2 Statistical results**

**3.2.1 Geographical distribution**

Firstly, the horizontal distribution characteristics of convective overshooting over East China are analysed by designing a more accurate algorithm for convective overshooting determination. Accurate determination of tropopause height is the first step of the convective overshooting determination algorithm. We first analyze geographical distribution of tropopause over East China calculated from ERA5, shown as Fig. 1b. In general, the tropopause height over East China is between 11.6 km and 16.7 km and has an obvious zonal distribution pattern: Tropopause height over SC and southern MEC (18-36 °N) is the highest and has small spatial variabilities, concentrated at ~16.7 km. Over northern MEC (36-38 °N), tropopause height obviously decreases and forms a gradient, which decreases to 16 km. Tropopause height over NC is the smallest and continues to decrease in a gradient pattern from south to north, decreasing to 13 km near central NC (45 °N) and 12 km near northern NC (48 °N). Minimum standard deviation of tropopause height appears in SC, along with central and southern MEC, lower than 0.2 km. From northern MEC to northern NC, the standard deviation first increases and then decreases, reaching a maximum of more than 2 km around 42 °N, and standard deviation over NC is generally above 1 km.

Obtaining storm top height from precipitation data is the second step of convective overshooting algorithm. Fig. 4 show geographical distribution of storm top height for total precipitation, convective precipitation and convective overshooting. As shown, mean storm top height over East China vary from 4.5 km to 8.5 km, while convective storm top height is mainly distributed between 3.5 km and 9 km. Convective storm top height over NC and northern MEC are the highest, with most areas exceeding 6.5 km and as we noted above, tropopause height in these two regions are lower (Fig. 1b), it can be inferred that convective overshooting events are more likely to occur. Further analysis of the frequency of convective overshooting in the following text will confirm this point. Compared with NC, convective storm top height over SC and southern MEC is lower, mainly distributed below 6.5 km. Storm top height of convective overshooting range from 10 km to 21 km (Fig. 4c), obviously much higher than normal precipitation (total and convective precipitation) and increasing gradually from north to south. Storm top height of convective overshooting over NC and northern MEC are low, distributed between 10 km and 16 km, which is because that their lower tropopause height (Fig. 1b) allow convection with lower storm





top height to penetrate troposphere, lowering the mean storm top height of convective overshooting,
while that over SC and southern MEC range from 16 km to 21 km, with higher tropopause height (Fig.
1b), allowing only stronger convection to penetrate the troposphere.
Based on the tropopause and storm top height information calculated above, algorithm for convective
overshooting determination over East China is designed and its geographical distribution of sample size
and frequency are shown as Table 1 and Fig. 5. The frequency of convective overshooting is defined as
the number of convective overshooting events divided by the total observed sample number of GPM
DPR. Statistical results indicate that the frequency of the convective overshooting over East China is
very low, with a magnitude of only $10^{-3}$, with regionally different. NC has the highest frequency of
convective overshooting, with sample size of 2394 (count, ct), followed by MEC with 582 ct, and SC is
the lowest (296 ct). Convective overshooting over NC and northern MEC, whose frequency range from
$4\times10^{-4}$ to $5.4\times10^{-3}$, occur more frequently than SC and southern MEC, whose frequency is between
$2\times10^{-4}$ and $6\times10^{-4}$, which is mainly because the former has a lower tropopause height and it's easier for
convective overshooting to occur.

### 3.2.2 Vertical structures

Based on the reflectivity profiles and the rain-rate profiles provided by the GPM DPR instrument, we
studied the vertical structure of precipitation within convective overshooting. DPDH (Distribution of
Probability Density with Height) analysis of radar reflectivity can effectively indicate the
three-dimensional structure characteristics of precipitation, which is therefore applied in a large number
of precipitation studies (Yuter and Houze, 1995). Fig. 6 shows DPDH of the DPR radar reflectivity. In
general, radar reflectivity within convective overshooting is obviously stronger and its storm top height
is higher. And the DPDH analysis also shows obviously regional differences. Radar echo intensity of
convective overshooting over NC is the weakest, and the echo near surface is mainly distributed from 25
dBZ to 55 dBZ, with sharp peak 47 dBZ, while the peak of the total precipitation is around 16 dBZ. And
the max radar echo top within convective overshooting over NC can reach to 13.5 km, 3.3 km higher than
the mean precipitation. Compared with NC, radar reflectivity within convective overshooting over SC
and MEC are stronger and their DPDH feature are more similar. Their echo top height is ~18 km, 6.5 km
higher than total precipitation, 4.5 km higher than NC, and their echo near surface concentrated around
30-55 dBZ, while that of total precipitation is between 15 dBZ and 43 dBZ. Besides, Radar reflectivity of



convective overshooting over NC accounts for a higher proportion below the zero level, while MEC and
SC account for a higher proportion above the zero level, which indicate that the upward motion within
convective overshooting over MEC and SC are stronger and there are more ice crystal particles.
Quantitative analysis of the vertical structure of precipitation within convective overshooting is one of
the main issues of interest to this study. Shown as Fig. 7, the rain rate profiles of convective overshooting
are provided, and to highlight its unique feature, rain rate profiles of total precipitation and convective
precipitation are also given. In general, the rain rate of convective overshooting is very higher, especially
below the zero level (~5 km), 5-10 times than normal precipitation, indicating stronger convection and a
greater precipitation of ice. In addition, differences between three regions are obvious. Rain rate of
convective overshooting over NC is about twice lower than MEC and SC, which is consistent with the
results of radar echo. At 1 km altitude, rain rate of convective overshooting are 12 mm/h (NC),22.5
mm/h (MEC), and 23 mm/h (SC) respectively. Below zero level, the variation of rain rate with altitude is
not very obvious, and difference of rain rate between convective overshooting and normal precipitation
are ~8 mm over NC and ~20 mm over MEC and SC. Above zero level, rain rate of convective
overshooting decreases obviously with altitude increasing, and rain rate are 6mm/h (NC), 10 mm (MEC)
and 6.5mm (SC) at 10 km. However, rain rate of other precipitation are no more than 2 mm/h above 8 km,
we therefore suggest that the strong upward flow within convective overshooting brings large amounts of
moisture from the lower layer to the upper layer.
We conduct the Probability Density Function (PDF) analysis on the Near Surface Rain Rate (NSRR)
within convective overshooting, and that of total and convective precipitation are also calculated, shown
as Fig. 8. Grade of precipitation are as follows: Light rain: <4.9 mm/12 h, Moderate rain: 5.0-14.9
mm/12h, Heavy rain: 15.0-29.9 mm/12h, Torrential rain: 30.0-69.9 mm/12h, Downpour: 70.0-139.9
mm/12h, and Heavy downpour: ≥140.0 mm/12h (General Administration of Quality Supervision, 2012).
The PDF curve of NSRR of convective overshooting is obviously different from normal precipitation,
and has regional differences. The peak value of PDF of convective overshooting appears at ~10 mm/h,
belonging to downpour, however, that of normal precipitation appear at ~1 mm/h, belonging to moderate
rain, which is obviously lower than convective overshooting. And the PDF of peak value of convective
overshooting over NC is about 11.5%, while that over MEC and SC are about 6%. Besides, sample size
of convective overshooting with precipitation grade of heavy downpour account for 34.0% (NC),46.7%
(MEC) and 34.8% (SC) respectively, 3-10 times than normal precipitation, which remind us to pay



special attention to the extreme precipitation events caused by convective overshooting that may cause
harm to our production and life.

### 3.2.3 Microphysical features

GPM center provides particle spectrum from dual-frequency radar. Based on the DSD profiles from
2ADPR, we further investigate the microphysical structures of convective overshooting. The Liquid
Water Path (LWP) and Ice Water Path (IWP) show the overall water content in the atmospheric column,
which is closely associated with microphysical processes within convective overshooting. To quantify
the characteristics of LWP and IWP within convective overshooting, the PDF of LWP and IWP of
convective overshooting are shown as Fig. 9, and that of convective and total precipitation are also
shown for comparison. The LWP and IWP within convective overshooting are the highest, with high
value of PDF mainly distributed around 1000 $g/m^3$ and 5000 $g/m^3$ respectively, much higher than that of
normal precipitation, which are around 100 $g/m^3$ and 300 $g/m^3$, indicating sufficient water vapor inside
convective overshooting. And differences of IWP between convective overshooting and normal
precipitation are bigger than LWP, suggesting that differences of water vapor above zero level between
them is greater and convective overshooting brings water vapor from bottom of the troposphere to higher
layers. Besides, differences of LWP and IWP between three regions are also worth noting: The LWP and
IWP over MEC and SC are more similar and higher than NC. Especially, LWP over MEC has a bimodal
structure with peaks of 630 and 5000 $g/m^3$, which are consistent with the bimodal structure of NSRR
PDF curve in Fig. 8. Analysis above in Fig. 1b shows that tropopause height over northern MEC is lower
than southern MEC, making convective overshooting easier happen over northern MEC, which indicates
that there are two types of convective overshooting events over MEC, weak events with lower storm top
height and strong events with higher storm top height, which correspond to the two peaks of LWP PDF
curve respectively.
We further use DSD parameter profiles, including the effective radius ($D_0$) and droplet concentration
($dBN_0$) profiles, to analyze the microphysical characteristics within convective overshooting, shown as
Fig. 10. Results show that the microphysical processes within convective overshooting are unique,
leading to various properties of the droplets in precipitation. Droplets of convective overshooting are
large, but sparse. Influenced by strong updrafts, precipitation particles within convective overshooting
continuously collide and grow large enough to fall, therefore, the effective radius of droplets are big,



below 10 km altitude, almost exceeding 2.5 mm, which is about twice than that of normal precipitation.
However, the droplet concentration within convective overshooting is relatively lower. Differences of
microphysical structure between three regions are also worth noting. Convective overshooting events
over NC have large, but sparse droplets, while that over SC have small, but dense droplets, and the
effective radius and concentration of droplets over MEC are between NC and SC, which is speculated
that it's related to the differences of aerosol content and types over three regions. Specifically, at 1 km
altitude, the effective radius of droplets over NC is the largest (2.87 mm), followed by MEC (2.7 mm),
and SC is the lowest (2.5 mm). As altitude increases, the effective radius of droplets first increase and
then decrease, with maximum of 2.93 mm over NC at 2.5 km and sharp peak over MEC (2.85 mm) and
SC (2.76 mm) near zero level, about twice than normal precipitation. The effective radius of droplets for
convective overshooting over NC and MEC are lower than 2.5 mm above 10 km and 12 km respectively.
It's worth noting that the effective radius of droplets for convective overshooting over SC show an
increasing trend above 8 km altitude, which are similar to convective precipitation, and their effective
radius of droplets over three regions also show an increasing tend from 9 km to 13 km, which may be
related to the strong upward motion inside. When the upward motion is strong, ice particles must grow
large enough to fall (Langmuir, 1948). Droplet concentration basically decreases with altitude, and that
within convective overshooting is obviously lower than normal precipitation and NC is the lowest, while
MEC and SC are higher and similar. Droplet concentration within convective overshooting near ground
is the highest, with NC (25.4), MEC (28) and SC (28), while that of normal precipitation is mainly
distributed between 32 and 35.
Convective overshooting plays an important role in exchanging constituents and energy between
troposphere and stratosphere. To quantify its impact, temperature, humidity, vertical velocity, and ozone
profiles from ERA5 are used to further analyze the internal thermodynamic characteristics of convective
overshooting and their impacts on atmospheric composition. The atmospheric temperature (Fig. 11aei)
and absolute humidity profiles within convective overshooting are statistically analyzed (Fig. 11bfj), and
the difference profiles between them with the total atmospheric profiles are shown as Fig.12ab. The
atmospheric temperature has a wavy response to convective overshooting, and the response varies in
different regions. Warming and cooling effect caused by convective overshooting over NC is the most
obvious compared with MEC and SC, no more than 1K. Near surface, convective overshooting exhibits
warming effect, that over NC is 2 K, and that over MEC and SC are no more than 1 K. From surface to ~2



km altitude, warming effect caused by convective overshooting over three regions gradually decreased.
From 5 km to the middle tropopause height, convective overshooting shows cooling effect over NC, with
sharp peak -4 K at 11.5 km altitude. Near tropopause, convective overshooting show warming effect over
NC and MEC, and the most warming effect over NC can reach 4 K at 16.5 km altitude; Convective
overshooting has an obvious humidifying effect on the air below the cloud top, with humidifying MEC
most and NC least. The humidification effect caused by convective overshooting first increases and then
decreases, with maximum of 2.3 g/m$^3$ (MEC), 1.45g/m$^3$ (SC) and 0.8 g/m$^3$ (NC) at 1 km altitude.
To further explore dynamic structure characteristics within convective overshooting and its effect on
ozone, we calculated the vertical velocity profiles (Fig. 11cgk) and ozone profiles (Fig. 11dhl) within
convective overshooting. And the difference profiles between them with the total atmospheric profiles
are shown as Fig.12cd. Results show that upward motion within convective overshooting is very strong,
the upward motion over MEC is the strongest, followed by SC and NC, and that over SC is slightly
higher than NC. From surface to middle tropopause, the upward motion within convective overshooting
firstly becomes stronger and then weakens. The vertical velocity over MEC can reach a maximum of
-0.85 Pa/s at 6 km altitude. The vertical velocity over SC is about -0.35 Pa/s from 6 km to 11.5 km. The
highest vertical velocity over NC is -0.3 Pa/s at 6.5 km. At the bottom of the stratosphere, there is a slight
downward movement of air, which is because the mixture of strong divergent flow and turbulence
mechanically drags the air above the clouds outward, and the air above the clouds are pulled down due to
continuity.
The impact of convective overshooting on ozone can be divided into three parts: Below 10 km (middle
and lower troposphere), convective overshooting makes ozone mass mixing ratio slightly decrease, with
decrease no more than 0.03 ×10$^{-6}$ kg/kg; From 10 km to 26 km (upper and middle troposphere and lower
stratosphere), convective overshooting obviously increases ozone mass mixing ratio. The increase of
ozone mass mixing ratio caused by convective overshooting over NC is the highest, with sharp peak
0.3 ×10$^{-6}$ kg/kg at 19 km, followed by MEC, with sharp peak 0.25 ×10$^{-6}$ kg/kg at 19 km, and SC is the
lowest, with sharp peak ~0.1 ×10$^{-6}$ kg/kg from 21 to 25 km; Above 26 km, ozone mass mixing ratio
decreases obviously caused by convective overshooting, with the maximum of 0.24 ×10$^{-6}$ kg/kg. The
most obvious change of ozone mass mixing ratio caused by convective overshooting is the increase of
ozone at tropopause and lower stratosphere, which is partly due to the strong upward motion within
convective overshooting (Fig. 12c) causing the influx of ozone from lower troposphere. On the other



hand, it's due to the sinking of air with high concentration ozone in the stratosphere, and the descent
vertical velocity at the bottom of the stratosphere also confirms this (Fig. 12c).
**4 Summary and conclusions**
The microphysical characteristics of convective overshooting are essential but poorly understood due to
the difficulty in accurately detecting the convective overshooting and obtaining microphysical
parameters during severe weather events. Based on the microphysical precipitation data from GPM DPR
and the meteorological data from ERA5 data, we designed a more accurate algorithm for convective
overshooting determination and examine the particle size, concentration, phase state and other
parameters of the convective overshooting over East China. The main conclusions are:
Firstly, the horizontal distribution characteristics of convective overshooting over East China are
analysed by designing a more accurate algorithm for convective overshooting determination. Statistical
results indicate that the frequency of the convective overshooting over East China is very low, with a
magnitude of only $10^{-3}$, with large regional differences. Convective overshooting events occur obviously
more frequently over NC and northern MEC, than SC and southern MEC, mainly because of the lower
tropopause height of the former and the different underlying surfaces. The mean convective overshooting
storm top height mostly ranges from 10 km to 21 km and has obvious regional distribution differences.
And convective overshooting storm top height over NC is 5-6 km higher than SC.
Based on the reflectivity profiles and the rain-rate profiles provided by the GPM DPR instrument, we
studied the vertical structure of precipitation within convective overshooting. The DPDH analysis of the
radar reflectivity shows that radar reflectivity within convective overshooting is obviously stronger and
its storm top height is higher. And the DPDH analysis also shows obviously regional differences. Radar
reflectivity of convective overshooting over NC accounts for a higher proportion below the zero level,
while MEC and SC account for a higher proportion above the zero level, which indicate that the upward
motion within convective overshooting over MEC and SC are stronger and there are more ice crystal
particles. Rain rate results also show that rain rate within convective overshooting is higher, 5-10 times
than that of normal precipitation. Especially, sample number of strong precipitation with grade of
precipitation of heavy downpour accounts for 34.0% (NC), 46.7% (MEC), and 34.8% (SC), which
remind us to pay special attention to the extreme precipitation events caused by convective overshooting.



GPM center provides particle spectrum from dual-frequency radar. Based on the DSD profiles from
2ADPR, we further investigated the microphysical structures of convective overshooting. Statistical
results show that convective overshooting has unique microphysical characteristics compared with
normal precipitation, with obvious regional differences. The LWP and IWP within convective
overshooting are abundant, with high values of PDF distributed around 1000 $g/m^3$ and 5000 $g/m^3$
respectively. Moreover, influenced by strong updrafts, precipitation particles within convective
overshooting continuously collide and grow large enough to fall, therefore, the effective radius is big,
below 10 km altitude, almost exceeding 2.5 mm, which is about twice than that of normal precipitation.
However, the droplet concentration within convective overshooting is relatively lower. Differences of
microphysical structure between three regions are also worth noting. The effective radius of droplet over
NC is slightly bigger than MEC and SC, while the droplet concentration is lower, which is speculated
that it's related to the differences of aerosol content and types over three regions.
Convective overshooting plays an important role in exchanging constituents and energy between
troposphere and stratosphere. To quantify its impact, temperature, humidity, vertical velocity, and ozone
profiles from ERA5 are used to further analyse the internal thermodynamic characteristics of convective
overshooting and their impacts on atmospheric composition. The atmospheric temperature has a wavy
response to convective overshooting, and the response varies in different regions. Convective
overshooting has an obvious humidifying effect on the atmosphere below the cloud top, with
humidifying MEC most and NC least. The upward motion within convective overshooting is very strong,
and the order of the ascending velocity between three regions corresponds to the order of their
humidifying effects, which also reflects that the humidifying effect of convective overshooting is related
to internal dynamics, consistent with study of Chae et al. (2011). In addition, convective overshooting
events not only bring ozone from lower troposphere to upper troposphere and lower stratosphere, but also
sink air with high concentration ozone in the stratosphere, thereby reducing the ozone in the stratosphere
and lower troposphere, and obviously increasing the ozone in the upper troposphere and lower
stratosphere.
Quantitative study of the internal microphysical characteristics within convective overshooting has not
been documented previously. Findings of this study may have important implications for the
microphysical evolution associated with convective overshooting, and provide more accurate
precipitation microphysical parameters as the input of the model simulation. This study is the



continuation of the previous research (Sun et al., 2021). In the future, we will further explore the impact
of aerosol on the internal microphysical characteristics within convective overshooting, and more
microphysical parameters with higher spatiotemporal resolution are expected to provide more detailed
features.
**Data availability.** ERA5 data are taken from
https://www.ecmwf.int/en/forecasts/datasets/reanalysis-datasets/era5. GPM DPR data are archived at
https://gpm.nasa.gov/data/directory.
**Acknowledgements.** This work was funded by the National Natural Science Foundation of China
Project (Grant No. 42230612) and the fellowship of China Postdoctoral Science Foundation (Grant
Numbers: 2022M723011).
**Author contributions.** Sun N., Lu G.P., and Fu Y.F. framed up this study. All the authors discussed the
concepts. Sun N. conducted the data analyses. Sun N. drafted the manuscript and all authors edited the
manuscript.
**Competing interests.** The authors declare no competing interests.

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





**Tables**

**Table1.** The sample number of total precipitation, convective precipitation, and convective overshooting over NC, MEC, and SC.

| Sample number (count, ct) | NC | MEC | SC |
|---|---|---|---|
| Total Precipitation | 652489 | 546313 | 319127 |
| Convective Precipitation | 111903 | 137674 | 111900 |
| Convective Overshooting | 2394 | 582 | 296 |





**Figures**

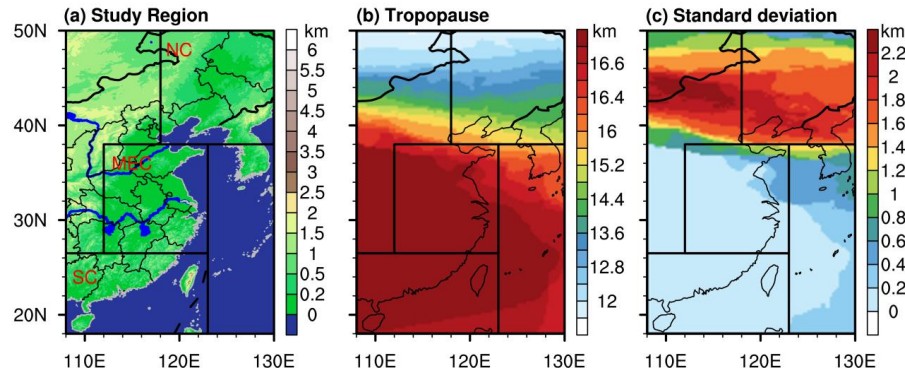


**Figure 1.** Study areas and their tropopause characteristics. **(a)** Regionalization of East China (Black boxes:
Divisions between NC, MEC and SC, and only the land surface is studied) and their terrain features. **(b)**
Distribution of tropopause height. **(c)** Distribution of standard deviation of tropopause height.

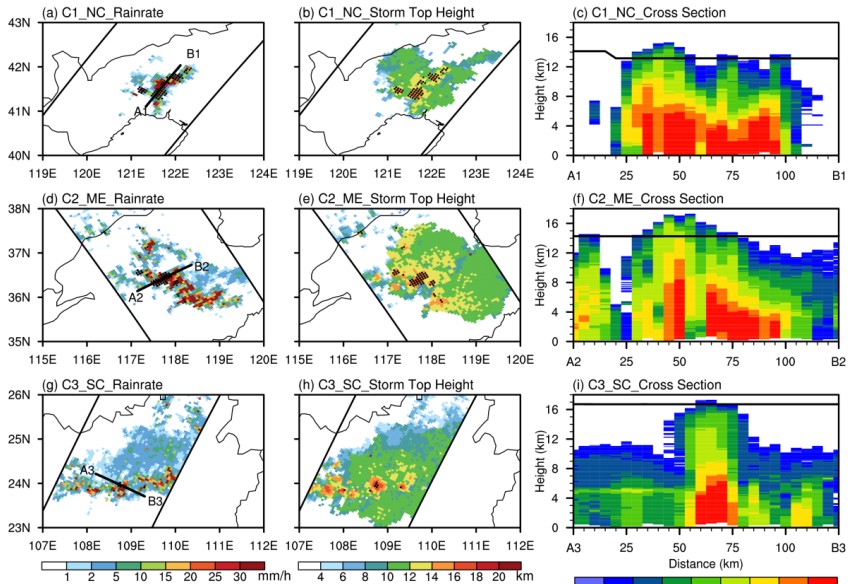

**Figure 2.** Precipitation characteristics of convective overshooting cases. **(a)** Distribution of rain rate of Case 1 (C1) and the occurrence time of C1 is at 14:00 on July 1, 2017. (The pixels in which convective overshooting occurs are marked as black boxes). **(b)** Distribution of storm top height of C1. **(c)** Radar reflectivity cross section along A1B1 and the black line show the tropopause height along A1B1. **(d)** Distribution of rain rate of C2 and the occurrence time of C2 is at 13:00 on July 30, 2015. **(e)** Distribution of storm top height of C2. **(f)** Radar reflectivity cross section along A2B2. **(g)** Distribution of rain rate of C3 and the occurrence time of C3 is at 17:00 on June 13, 2015. **(h)** Distribution of storm top height of C3. **(i)** Radar reflectivity cross section along A3B3.

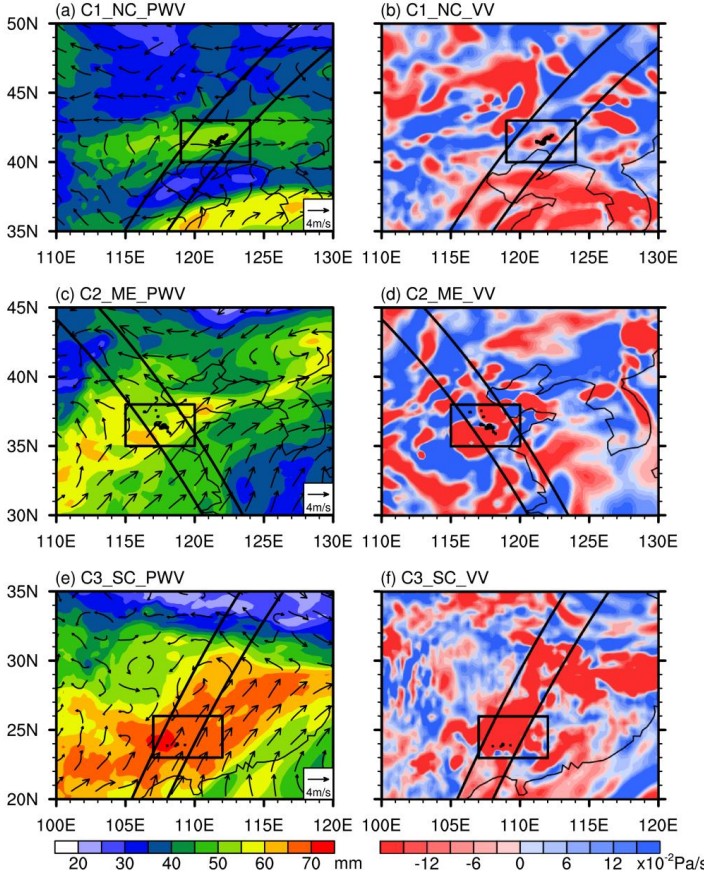

**Figure 3.** Characteristics of large scale circulation of convective overshooting cases. **(a)** Distribution of precipitable water vapor (PWV) and streamlines at 850 hPa of C1. The area where the case occurred is marked as big black boxes and the pixels in which convective overshooting occurs are marked as little black boxes. The black line is the GPM detection orbit. **(b)** Distribution of vertical velocity (VV) at 500 hPa of C1. **(c)** Distribution of PWV and streamlines of C2. **(d)** Distribution of VV of C2. **(e)** Distribution of PWV and streamlines of C3. **(f)** Distribution of VV of C3.



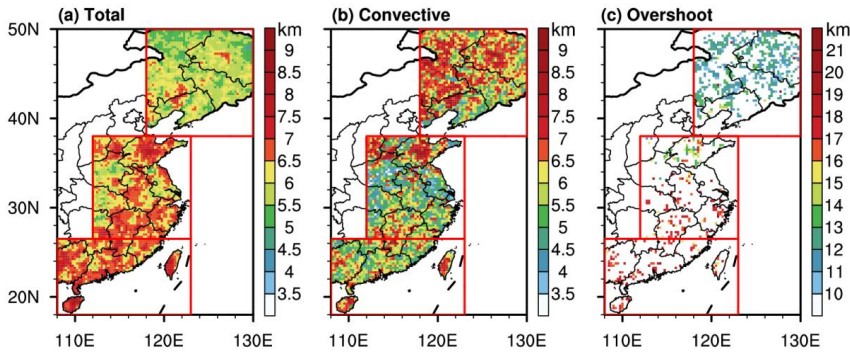

**Figure 4.** Geographical distribution of storm top height. **(a)** Distribution of storm top height for total precipitation. **(b)** Distribution of storm top height for convective precipitation. **(c)** Distribution of storm top height for convective overshooting.



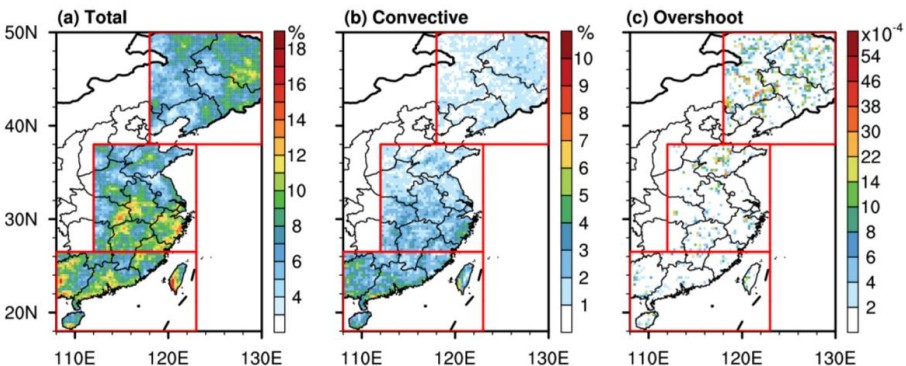

**Figure 5.** Precipitation frequency. **(a)** Frequency of total precipitation. **(b)** Frequency of convective precipitation.
**(c)** Frequency of convective overshooting.

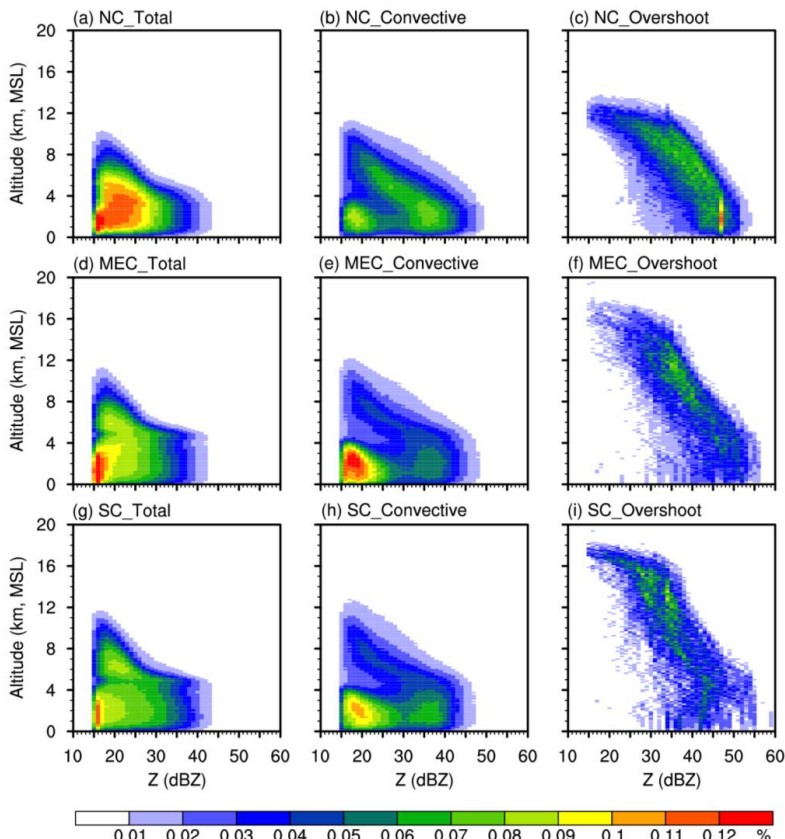

**Figure 6.** DPDH (Distribution of Probability Density with Height) of radar reflectivity. **(a)** DPDH for total precipitation over NC. **(b)** DPDH for convective precipitation over NC. **(c)** DPDH for convective overshooting over NC. **(d)** DPDH for total precipitation over MEC. **(e)** DPDH for convective precipitation over MEC. **(f)** DPDH for convective overshooting over MEC. **(g)** DPDH for total precipitation over SC. **(h)** DPDH for convective precipitation over SC. **(i)** DPDH for convective overshooting over SC.



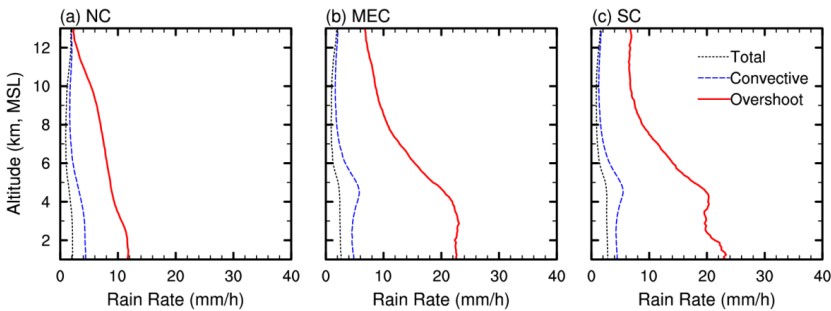

**Figure 7.** Rain rate profiles for total precipitation, convective precipitation and convective overshooting (Red lines are convective overshooting; Blue lines are the convective precipitation; Black lines are the total precipitation). **(a)** The rain rate profiles over NC. **(b)** The rain rate profiles over MEC. **(c)** The rain rate profiles over SC.




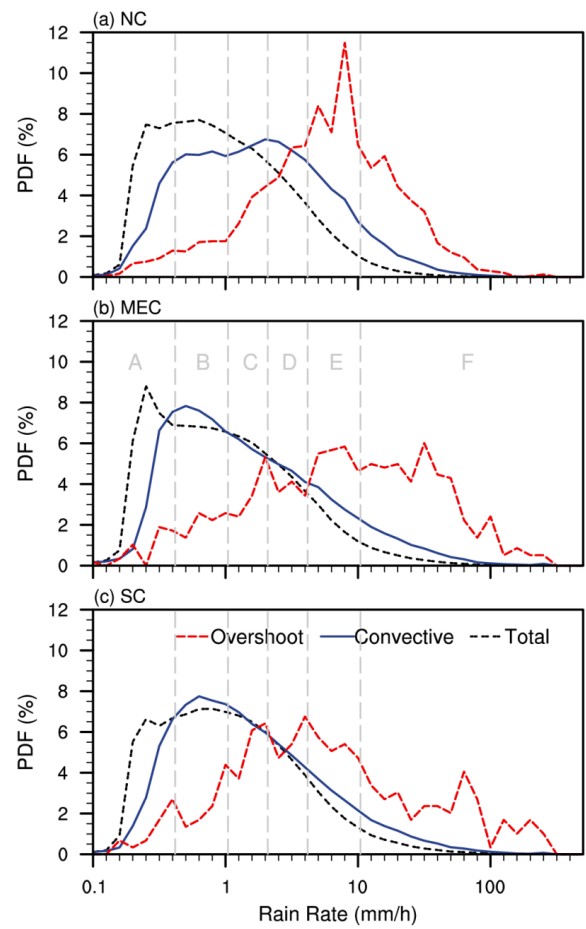

**Figure 8.** Probability Density Function (PDF) of Near Surface Rain Rate (NSRR). **(a)** PDF of NSRR in NC. **(b)** PDF of NSRR in MEC. **(c)** PDF of NSRR in SC.



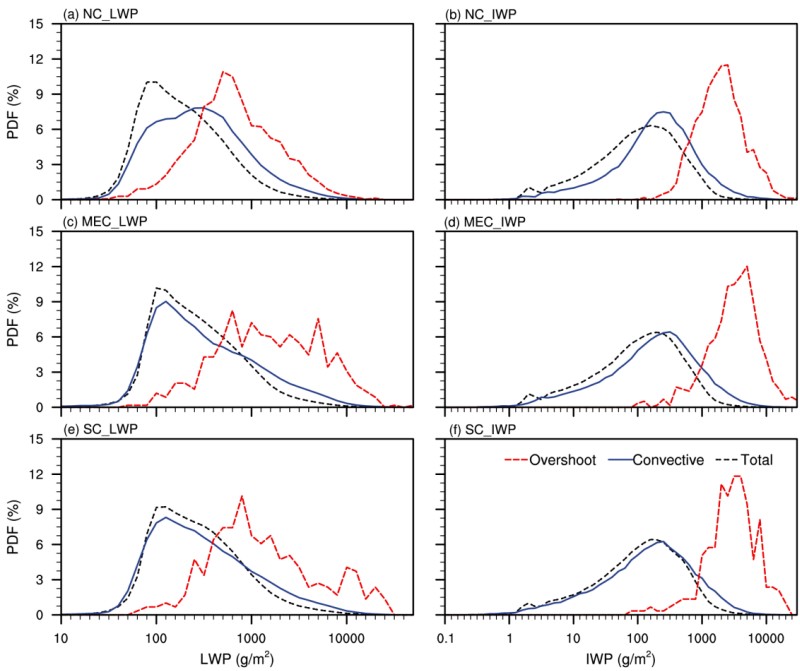

**Figure 9.** PDF of Liquid Water Path (LWP) and Ice Water Path (IWP). **(a)** PDF of LWP over NC. **(b)** PDF of IWP over NC. **(c)** PDF of LWP over MEC. **(d)** PDF of IWP over MEC. **(e)** PDF of LWP over SC. **(f)** PDF of IWP over SC.



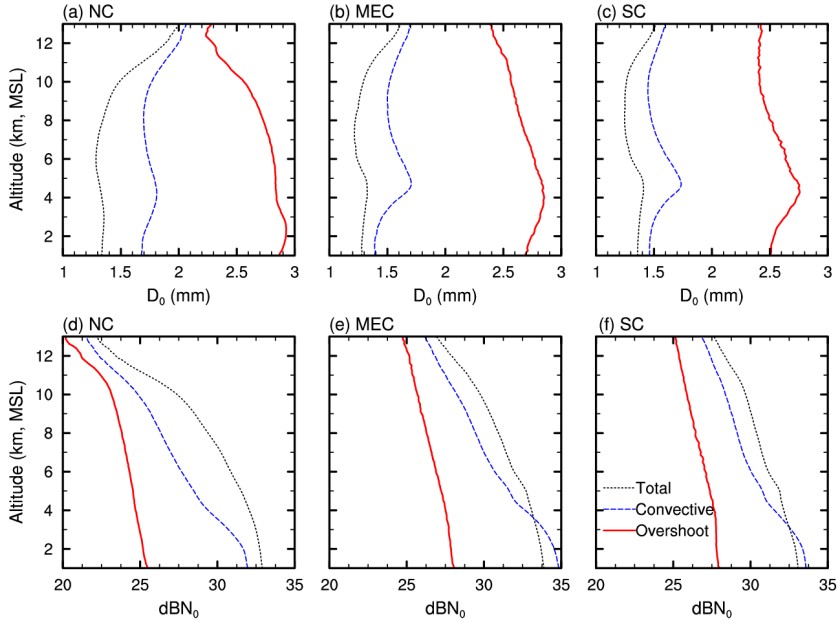

**Figure 10.** The droplet concentration ($dBN_0$) and effective radius ($D_0$) profiles for total precipitation, convective precipitation and convective overshooting over NC, MEC and SC. **(a)** The $dBN_0$ profiles over NC. **(b)** The $dBN_0$ profiles over MEC. **(c)** The $dBN_0$ profiles over SC. **(d)** $D_0$ profiles over NC. **(e)** $D_0$ profiles over MEC. **(f)** $D_0$ profiles over SC.

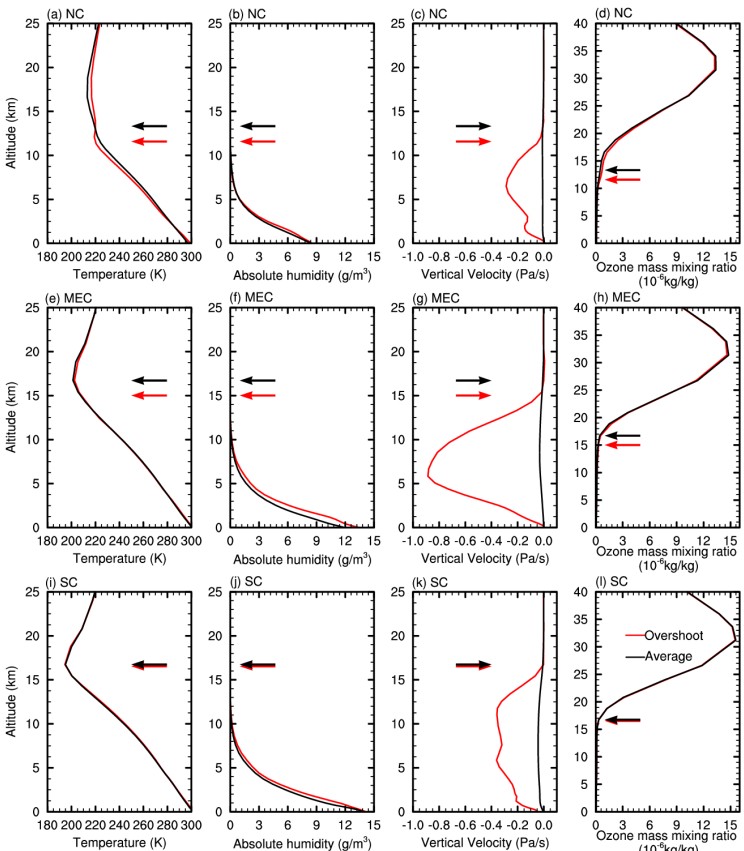

**Figure 11.** Atmospheric parameter profiles for total precipitation and convective overshooting over NC, MEC and SC. **(a)** Temperature profiles over NC. **(b)** Absolute humidity profiles over NC. **(c)** Vertical velocity profiles over NC. **(d)** Ozone mass mixing ratio profiles over NC. **(e)** Temperature profiles over MEC. **(f)** Absolute humidity profiles over MEC. **(g)** Vertical velocity profiles over MEC. **(h)** Ozone mass mixing ratio profiles over MEC. **(i)** Temperature profiles over SC. **(j)** Absolute humidity profiles over SC. **(k)** Vertical velocity profiles over SC. **(l)** Ozone mass mixing ratio profiles over SC.





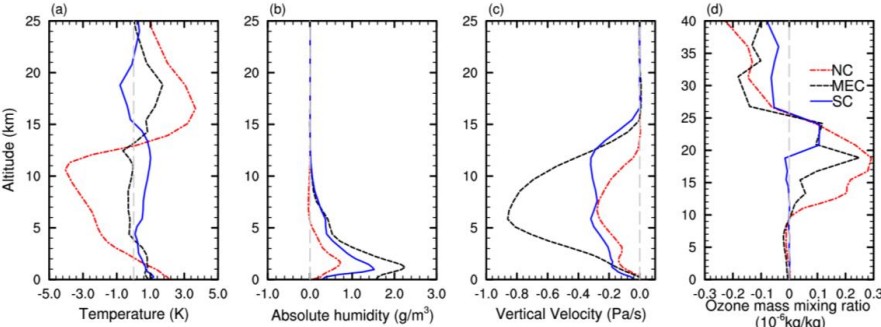

**Figure 12.** Difference of atmospheric parameters profiles between total precipitation and convective overshooting over NC, MEC and SC. **(a)** Difference of temperature profiles. **(b)** Difference of absolute humidity profiles. **(c)** Difference vertical velocity profiles.