# Peer review of "Microphysical characteristics of precipitation within"

_EGUsphere, 2023_

## Author Comment (AC1)

**Dear Referee,**

    **I am very impressed by your so carefully checking and revising the manuscript. Thank you so much! I have carefully read all your question and suggestion, and modifications have been made in the manuscript. My replies are as follows.**

**General Comments**

Overall, this paper contributes to an active and important area of study by investigating the microphysical characteristics of overshooting convection over East China. The techniques employed are reasonable and the results will be valuable for ongoing research in the area. However, the paper would benefit from a more detailed description of the methods, and various grammatical errors impact the flow and readability. Therefore, I suggest the following changes be made to the paper before publication.

**Specific Comments**

Introduction

(1) You mention the cold point and WMO tropopause definitions, but there is also the Dynamic Tropopause, which is based on the differing values of potential vorticity in the troposphere and stratosphere. I agree that the WMO tropopause definition is most suitable for identifying overshooting convection, but I might mention that there are three widely used definitions, rather than two.

Answer: Thanks for your advice! Dynamic tropopause has been added in this manuscript, meanwhile, I also find another new tropopause definition: ozone tropopause. And these two definitions have been added in the manuscript, shown as line 95-102 in the manuscript. The added references are shown as follows.

Danielsen, E. F., Hipskind, R. S. and Gaines, S. E. et al.: Three-dimensional analysis of potential vorticity associated with tropopause folds and observed variations of ozone and carbon monoxide, Journal of Geophysical Research: Atmospheres, 92(D2), 2103-2111, https://doi.org/10.1029/JD092iD02p02103, 1987.

Holton, J. R., Haynes, P. H. and McIntyre, M. E. et al.: Stratosphere-troposphere exchange. Reviews of geophysics, 33(4), 403-439, https://doi.org/10.1029/95RG02097, 1995.

Bethan, S., Vaughan, G., and Reid, S. J.: A comparison of ozone and thermal tropopause heights and the impact of tropopause definition on quantifying the ozone content of the troposphere, Quarterly Journal of the Royal Meteorological Society, 122(532), 929-944, https://doi.org/10.1002/qj.49712253207, 1996.

Zahn, A., Brenninkmeijer, C. A. M., and Van Velthoven, P. F. J.: Passenger aircraft project CARIBIC 1997–2002, Part I: the extratropical chemical tropopause,

Atmospheric Chemistry and Physics Discussions, 4(1), 1091-1117, https://doi.org/10.5194/acpd-4-1091-2004, 2004.

Data and Methods

(2) What is the reasoning behind the placement of the three regions? You mention their different climatic characteristics, but it would be good to be more specific about what these differences are. It would also be good to explain why you have limited the study area to the land.

Answer:Thanks for your question and advice!

1. Placement of the three regions is based on the study of Xia (2015). Using years of NCEP/NCAR reanalysis data, Xia (2015) analyzed the climatic feature of temperature and water vapor in China and divided China into different climatic zones. And we divided East China into three climatic zones according to Xia's study (2015). And this explanation has been added in the manuscript, shown as line 161-165.

2. For the three regions, the lower latitude areas have higher surface temperature, greater temperature lapse rate and lower temperature of stratosphere. Temperature profiles of same latitude are essentially same over SC and MEC, and temperature signals exist meridional differences over NC. Atmospheric humidity has remarkable regional characteristics, SC is wetter, with the surface relative humidity of more than 70%, while NC and MEC are drier and their humidity range from 50% to 70%. And this explanation has been added in the manuscript, shown as line 168-173.

3. Characteristics of vertical structure of precipitation over land and sea are very different. We have compared rain rate profiles between land and sea over SC, shown as Fig. 1, we can see that rain rate of convective overshooting over sea is much higher than land. This manuscript is limited in space and focuses only on the land region, a comparative study of convective overshooting over sea and land will be carried out in the future. And this explanation has been added in the manuscript, shown as line 158.

[Figure]

[Figure]

Figure 1 Rain rate profiles of Rain rate profiles for total precipitation, convective precipitation and convective overshooting over land and sea of SC. (a) The rain rate profiles over land of SC. (b) The rain rate profiles over sea of SC.

Xia, J.: Research on climatic regionalization of China and characteristics of temperature, humidity and wind in precipitation cloud, University of Science and Technology of China, 2015.

(3) Are you interpolating the ERA5 temperature profiles onto an altitude grid in order to calculate the tropopause height? The exact methods used to calculate the tropopause need to be explained in order for the results to be reproducible.

Answer:Thanks for your question and advice!

1. We are not use the method of interpolation. We use the principle of the nearest method to match the each pixels of GPM detection with ERA5 grid data. And this explanation has been added in the manuscript, shown as line 145.

2. The exact methods used to calculate the tropopause has been explained and shown as follows. And this explanation has been added in the manuscript, shown as line 145-155.

Firstly, match the each pixels of GPM detection with ERA5 grid data by using the principle of the nearest method. The marching time between GPM and ERA5 is 1 h, and the matching range is $0.25\,° \times 0.25\,°$. Storm top height is obtained from the GPM DPR. Convective overshooting is defined to occur where the storm top height is above the real-time tropopause height in a precipitation pixel.

Real-time tropopause height is calculated from the temperature profiles from ERA5 according to the definition from the World Meteorological Organization (1957). The algorithmic process is shown as follows: First, find X layer whose atmospheric lapse rate is 2 K $km^{-1}$ or less by judging start from the first layer (near the ground) of the temperature profile, and then judge whether the atmospheric lapse rate does not exceed 2 K $km^{-1}$ between the X level and all higher levels within 2 km, if so, the height of X layer is the tropopause height, if not, repeat the above algorithm starting from the X layer until tropopause layer is found.

Results

(4) Instead of saying 'rain rates are mostly over 20 mm/h', as in line 143, it would be better to provide a more quantitative result. You could calculate the mean rain rate for overshooting pixels, for example. This applies to the rain rates and storm top heights for the three case studies.

Answer: Thanks for your advice! The mean rain rate and storm top height for the overshooting pixels of the three case studies have been calculated and Modifications have been made in the manuscript, shown as line 181, 196, 191.

(5) In Figure 1b, I assume this is a climatological mean of the tropopause height? This should be stated for this and similar figures. Captions should include the time period over which these means are taken, which appears to be mentioned only in section 2.1. I would recommend specifying the study period in section 2.4 instead or in addition.

Answer: Thanks for your advice! Yes, Figure 1b show the climatological mean of the tropopause height, and we have stated that in captions of Figure 1b and in the text, shown as line 674-679. Time period was added in captions and section 2.4, and time period in section 2.1 was deleted, shown as line 174.

(6) In Figure 4, how is the distinction made between total precipitation and convective precipitation? Are these data sets complementary, or (for example) does the convective precipitation plot include values from overshooting pixels?

Answer: Thanks for your question!

1.The total precipitation represent the all pixels with rain rate higher than 0 mm/h detected by GPM DPR, and those pixels whose rain type are "Convective" are defined as convective precipitation. The detailed definition of total precipitation and convective precipitation are added in the manuscript, shown as line 227-229.

2. Convective precipitation plot include values from overshooting pixels.

**Technical Corrections**

Abstract

(7) 8 "We examine the geographical distribution and microphysical three-dimensional structure of convective overshooting over East China by comparing Global Precipitation …"

Answer: Thanks for your advice! Modifications have been made in the manuscript, shown as line 8.

(8) 13 "with a magnitude of only 10^-3;" This needs to be explained. It is not clear what it means without reading the paper. The semicolon most likely should be a period.

Answer: Thanks for your advice! Those words have been modified to "and its frequency varies from $4 \times 10^{-4}$ to $5.4 \times 10^{-3}$" in the manuscript, shown as line 14.

(9) 14 "below the zero level" Recommend changing "zero level" to "freezing level" here and elsewhere.

Answer: Thanks for your advice! Modifications have been made in the manuscript, shown as line 15, 16, 271, 272, 278, 283, 285, 315, 340, 412, 413.

(10) 14 "SC (South China)" Recommend changing to "South China (SC)" for consistency with earlier abbreviation definitions.

Answer: Thanks for your advice! All abbreviation definitions have been changed for consistency. Modifications have been made in the manuscript, shown as line 12, 15, 129, 166, 257, 757.

(11) 15 Semicolon should be a period.

Answer: Thanks for your advice! Modifications have been made in the manuscript, shown as line 14, 17, 364.

(12) 17 "Droplets of convective overshooting are large, but sparse, with an effective droplet radius of nearly 2.5 mm below 10 km, which is about twice that of non-overshooting precipitation."

Answer: Thanks for your advice! Modifications have been made in the manuscript, shown as line 18.

(13) 19 "humidifies air below the cloud top and increases ozone concentrations near the tropopause as a result of an influx of ozone from the lower troposphere and subsidence of high-ozone stratospheric air."

Answer: Thanks for your advice! Modifications have been made in the manuscript, shown as line 22.

(14) 23 "as input for model simulations."

Answer: Thanks for your advice! Modifications have been made in the manuscript, shown as line 28.

Introduction

(15) 38 "the effects of convective overshooting on the temperature of the UTLS has…"

Answer: Thanks for your advice! Modifications have been made in the manuscript, shown as line 44.

(16) 41 "at the Earth's surface, with important social and economic impacts"

Answer: Thanks for your advice! Modifications have been made in the manuscript, shown as line 49.

(17) 43 "impacts, it is of high importance"

Answer: Thanks for your advice! Modifications have been made in the manuscript, shown as line 50.

(18) 44 "overshooting, which have attracted considerable"

Answer: Thanks for your advice! Modifications have been made in the manuscript, shown as line 51.

(19) 49 "efficiency of water vapor transport to the lower stratosphere"

Answer: Thanks for your advice! Modifications have been made in the manuscript, shown as line 56.

(20) 54 "overshooting are larger than"

Answer: Thanks for your advice! Modifications have been made in the manuscript, shown as line 61.

(21) 55 "characteristics of convective overshooting"

Answer: Thanks for your advice! Modifications have been made in the manuscript, shown as line 62.

(22) 71 "that is to find pixels"

Answer: Thanks for your advice! Modifications have been made in the manuscript, shown as line 78.

(23) 72 "which improves the"

Answer: Thanks for your advice! Modifications have been made in the manuscript, shown as line 79.

(24) 81 "6-hourly dataset"

Answer: Thanks for your advice! Modifications have been made in the manuscript, shown as line 88.

(25) 82 "geographical distribution; the microphysical"

Answer: Thanks for your advice! Modifications have been made in the manuscript, shown as line 89.

Data and Methods

(26) 125 "2.3 Definition of convective overshooting" (remove 'the')

Answer: Thanks for your advice! Modifications have been made in the manuscript, shown as line 138.

(27) 126 "Convective overshooting is defined to occur where the storm top height is above"

Answer: Thanks for your advice! Modifications have been made in the manuscript, shown as line 147.

(28) 127 "Storm top height" (capitalize beginning of sentence)

Answer: Thanks for your advice! Modifications have been made in the manuscript, shown as line 147.

(29) 129 "as follows:"

Answer: Thanks for your advice! These words have been deleted due to the rewriting of this part.

Results

(30) Figure 2: The black boxes used to indicate overshooting almost entirely cover the gridbox, making it difficult to read the intended information from the plots (rainrate, etc.). I would also state when the cases occurred in the main text, rather than the caption here.

Answer: Thanks for your advice! The black boxes are reduced to make the information about rain rate and storm top height clear, shown as Figure 2. The occurrence time of cases in the caption are deleted, and the time are added in the main text, shown as line 179-189, 683-690.

 (31) You should indicate in the text that the locations of the three cases are shown on Figure 3.

Answer: Thanks for your advice! Modifications have been made in the manuscript, shown as line 196.

(32) 142 "Convective overshooting is observed in a total of 65 pixels for C1. Most overshooting pixels have rain rates exceeding 20 mm/h (Fig. 21), and storm top heights exceeding 12 km (Fig. 2b)." similar for case 2 (line 146) and case 3 (line 150)

Answer: Thanks for your advice! Modifications have been made in the manuscript, shown as line 179-190.

(33) 154 "characteristics of the large scale circulation for these three cases, we"

Answer: Thanks for your advice! Modifications have been made in the manuscript, shown as line 194.

(34) 155 "shown in Fig. 3."

Answer: Thanks for your advice! Modifications have been made in the manuscript, shown as line 196.

(35) 156 "In general, areas in which convective overshooting occur have abundant"

Answer: Thanks for your advice! Modifications have been made in the manuscript, shown as line 196.

(36) 159 "The PWV of the region in which overshooting occurs is between 50 and 55 mm, which is higher than elsewhere (Fig. 3a)"

Answer: Thanks for your advice! Modifications have been made in the manuscript, shown as line 199.

(37) 160 "Upward motion near the convective overshooting is strong, ranging from -0.03 to -0.12 Pa/s"

Answer: Thanks for your advice! Modifications have been made in the manuscript, shown as line 201.

(38) 176 "tropopause height decreases and forms"

Answer: Thanks for your advice! Modifications have been made in the manuscript, shown as line 218.

(39) 177 "height over NC is the lowest and"

Answer: Thanks for your advice! Modifications have been made in the manuscript, shown as line 219.

(40) 185 "East China varies from"

Answer: Thanks for your advice! Modifications have been made in the manuscript, shown as line 229.

(41) 192 "ranges from 10 km to 21 km (Fig. 4c), much higher than"

Answer: Thanks for your advice! Modifications have been made in the manuscript, shown as line 236.

(42) 194 "Storm top heights of convective"

Answer: Thanks for your advice! Modifications have been made in the manuscript, shown as line 238.

(43) 195 "which is due to a lower tropopause height (Fig. 1b) allowing convection with lower storm top height to penetrate the tropopause. This lowers the mean storm top height of convective overshooting in these regions, while tropopause heights over SC and southern MEC range from 16 km to 21 km (Fig. 1b), allowing only strong convection to penetrate the tropopause"

Answer: Thanks for your advice! Modifications have been made in the manuscript, shown as line 239-243.

(44) 199 "above, an algorithm"

Answer: Thanks for your advice! Modifications have been made in the manuscript, shown as line 244.

(45) 204 "with regional variation."

Answer: Thanks for your advice! Modifications have been made in the manuscript, shown as line 249.

(46) 216 "overshooting is stronger and"

Answer: Thanks for your advice! Modifications have been made in the manuscript, shown as line 262.

(47) 217 "also shows regional differences."

Answer: Thanks for your advice! Modifications have been made in the manuscript, shown as line 263.

(48) 231 "overshooting is much higher,"

Answer: Thanks for your advice! Modifications have been made in the manuscript, shown as line 277.

(49) 232 "5-10 times that of normal precipitation. This indicates stronger convection and a greater concentration of ice."

Answer: Thanks for your advice! Modifications have been made in the manuscript, shown as line 277.

(50) 234 "Rain rates of convective overshooting over NC are about half as high as over MEC and SC"

Answer: Thanks for your advice! Modifications have been made in the manuscript, shown as line 280.

(51) 239 "overshooting clearly decreases with increasing altitude, and rain rates are"

Answer: Thanks for your advice! Modifications have been made in the manuscript, shown as line 286.

(52) 240 "rain rates of"

Answer: Thanks for your advice! Modifications have been made in the manuscript, shown as line 287

(53) 248 "overshooting is clearly different"

Answer: Thanks for your advice! Modifications have been made in the manuscript, shown as line 295.

(54) 250 "classified as downpour, while that of normal precipitation appears at ~1 mm/h, classified as moderate rain."

Answer: Thanks for your advice! Modifications have been made in the manuscript, shown as line 297.

(55) 274 "making it easier for convective overshooting to occur over northern MEC. This indicates that"

Answer: Thanks for your advice! Modifications have been made in the manuscript, shown as line 321.

(56) 318 "has a humidifying effect on the air below the cloud top, humidifying MEC"

Answer: Thanks for your advice! Modifications have been made in the manuscript, shown as line 365.

(57) 336 "overshooting increases ozone"

Answer: Thanks for your advice! Modifications have been made in the manuscript, shown as line 383.

(58) 340 "decreases due to convective overshooting"

Answer: Thanks for your advice! Modifications have been made in the manuscript, shown as line 387.

Summary and Conclusions

(59) 356 "events occur more frequently"

Answer: Thanks for your advice! Modifications have been made in the manuscript, shown as line 403.

(60) 363 "is stronger"

Answer: Thanks for your advice! Modifications have been made in the manuscript, shown as line 410.

(61) 364 "also shows regional"

Answer: Thanks for your advice! Modifications have been made in the manuscript, shown as line 411.

(62) 389 "cloud top, humidifying MEC"

Answer: Thanks for your advice! Modifications have been made in the manuscript, shown as line 437.

(63) 396 "and increase the ozone"

Answer: Thanks for your advice! Modifications have been made in the manuscript, shown as line 444.

---

## Author Comment (AC2)

**Dear Referee,**

    **Thank you so much for reading the manuscript so carefully and providing so many valuable suggestions. We have learned a lot from your comments! Thanks again! We have carefully read all your question and suggestion, and modifications have been made in the manuscript. My replies are as follows.**

**General Comments**

(1) The authors refer to the temperature lapse-rate tropopause as the "thermodynamic" tropopause. It should instead by referred to as the "thermal" tropopause throughout. In addition, there are several alternative instances outlined under the specific comments section below of inappropriate, inaccurate, or unjustified claims in the text.

Answer: Thanks for your advice! Modifications have been made in the manuscript, shown as line 110, 117, 118.

(2) The motivation to carry out the study is principally focused on improving understanding of the microphysical characteristics of overshooting convection. The background bases this motivation on the need to clarify the efficiency of water vapor transported to the lower stratosphere by convective overshooting. However, the detailed analysis of the microphysical characteristics largely ignores characteristics near and within the overshoots. Rather, the focus is on altitudes at and below 12 km, which lie below the lowest tropopause altitudes over the analysis domain. The results presented are largely uninteresting and unsurprising given the modes evaluated (all observations, convection observations, and overshooting convection observations). The overshooting convection observations represent the extremes in convective depths, which (as expected) result in the highest liquid water paths and ice water paths. Conversely, the authors miss an opportunity to evaluate and contrast the characteristics specifically within the overshoots as more directly motivated in the analysis. Thus, I believe a more valuable contribution would be to revise the analysis to focus specifically on characteristics within the overshoot. To do so, it will be important to aggregate the data in a tropopause-relative altitude coordinate.

Answer: Thanks for your comments! 'the efficiency of water vapor transported to the lower stratosphere by convective overshooting' is really important and hot topic for 'improving understanding of the microphysical characteristics of overshooting convection'. However, the motivation of this manuscript is mainly focused on the vertical and microphysical structure of precipitation within the convective overshooting. Driven by this purpose, we use precipitation parameters including particle size, concentration, phase state and other parameters provided by GPM to deeply and comprehensively examine the precipitation structure within the convective overshooting. Therefore, water vapor transported by convective overshooting is not

the focus of this manuscript. In the future, we will combine multi-source data and modeling to further conduct detailed research on water vapor characteristics within convective overshooting.

As for 'the focus is on altitudes at and below 12 km', on the one hand, that is caused by the limited detection by GPM and detection above 12 km becomes unstable and the credibility of the data decreases. Therefore, we mainly use data below 12 km. On the other hand, for the study of water vapor transported by convective overshooting, study near the tropopause is more meaningful, but for the study of precipitation structure, we can see that the values of precipitation parameters above 12 km are very small, and the high value areas are mostly distributed below 12 km. From this point of view, it's still meaningful for focus precipitation parameters on altitudes at and below 12 km.

In summary, main purpose of this manuscript is not to study the impact of convective overshooting on water vapor, but to reveal the vertical and microphysical structure of precipitation within the convective overshooting, which is a gap in previous research, and the results of this manuscript can also provide more accurate precipitation microphysical parameters as input for model simulations.

(3) The use of ERA5 to diagnose anomalies in ozone and water vapor concentration for the events seems problematic. For one, ERA5 is not demonstrated to resolve well the overshooting process (detailed comparisons of overshoot occurrence/frequency with the GPM data would be a good way to solve that, but my guess is that it can't be shown convincingly on the model grid). Moreover, the horizontal and vertical resolution of ERA5 output is a considerable constraint on the degree to which meaningful results toward the study's goals can be obtained. Beyond convection, resolution impacts the extent to which changes in the environment can be reliably deduced. Finally, it is not clear to what extent ERA5 data are validated against observed composition and demonstrated to be reliable. For example, most reanalyses are far too wet in the upper troposphere and lower stratosphere. Thus, is there really any considerable value about the impacts of overshooting that can be gained from analyzing this output? The use of this data and study design do not provide compelling or convincing evidence to support that.

Answer: Thanks for your comments! As you suggested in specific comments, we have deleted that part. However, comparing ERA5 with other popular data, advantage of ERA5 is obvious, and we still believe that water vapor and temperature from ERA5 can be used in convective overshooting. Focus of this manuscript should be more on the discussion of precipitation structure, and analysis of this part of profiles from ERA5 are rough, so we delete this part.

At present, the most common methods for detecting water vapor include sounding detection, occultation detection and reanalysis data. Sounding detection is the most accurate method as it involves on-site exploration. We have compared water vapor

from ERA5, sounding detection (IGRA) and occultation detection (COSMIC), and results show that water vapor from ERA5 is relatively reliable (Sun et al., 2022), shown as Fig. 1. IGRA is the sounding detection, which can be used as a benchmark. Both case study and statistical results show that difference of water vapor between ERA5 and IGRA is small in the upper troposphere, indicating the credibility of water vapor from ERA5. Due to the lack of observation of IGRA near tropopause and lower stratosphere, we can only compare ERA5 with COSMIC. At this point, we can also see that although sounding data is more correct, it has obvious limitations in terms of detection height. Previous study has shown that water vapor from COSMIC is biased towards humidity (Kursinski et al., 1997). We can see that water vapor of ERA5 is generally lower than that of COSMIC near tropopause and lower stratosphere, indicating that ERA5 is relatively accurate compared to COSMIC. In addition, ERA5 has the highest spatiotemporal resolution, compared with other popular reanalysis data, such as JRA55 and MERRA2. In general, using ERA5 to study the impact of convective overshooting on temperature and water vapor is not a bad choice. In the future, we will refer to your suggestions and combine model simulation to conduct more detailed and in-depth research specifically on water vapor in the UTLS region.

Kursinski, E. R., Hajj, G. A., Schofield, J. T., Linfield, R. P., and Hardy, K. R.: Observing Earth's atmosphere with radio occultation measurements using the Global Positioning System. Journal of Geophysical Research: Atmospheres, 102(D19), 23429-23465, https://doi.org/10.1029/97JD01569, 1997.

Sun, N., Zhong, L., Zhao, C., Ma, M., and Fu, Y.: Temperature, water vapor and tropopause characteristics over the Tibetan Plateau in summer based on the COSMIC, ERA-5 and IGRA datasets, Atmospheric Research, 266, 105955, https://doi.org/10.1016/j.atmosres.2021.105955, 2022.

[Figure]

Figure 1 Case study and statistical study of water vapor profiles from COSMIC, ERA5, and IGRA

**Specific Comments**

(4) Lines 32-34: previous studies do not show that overshooting has a net dehydrating effect on the stratosphere. Several studies do show that convection and dehydrate the upper troposphere in the tropics, but otherwise convection has been universally shown to hydrate the stratosphere.

Answer: Thanks for your reminder, and modifications have been made in the introduction , shown as line 37-40.

(5) Line 39: The studies cited in this paragraph are almost entirely focused on tropical overshooting convection. Equal consideration/discussion related to prior work on midlatitude overshooting convection should be given here.

Answer: Thanks for your advice, and modifications have been made in the introduction, shown as line 45-50. And analysis of following references about midlatitude overshooting convection have been added in the introduction.

Smith, J. B., Wilmouth, D. M., and Bedka, K. M. et al.: A case study of convectively sourced water vapor observed in the overworld stratosphere over the United States, Journal of Geophysical Research: Atmospheres, 122(17), 9529-9554, https://doi.org/10.1002/2017JD026831, 2017.

Werner, F., Schwartz, M. J., and Livesey, N. J. et al.: Extreme outliers in lower stratospheric water vapor over North America observed by MLS: Relation to overshooting convection diagnosed from colocated Aqua‐MODIS data, Geophysical Research Letters, 47(24), e2020GL090131, https://doi.org/10.1029/2020GL090131, 2020.

Wang, X., Huang, Y., and Qu, Z. et al.: Convectively Transported Water Vapor Plumes in the Midlatitude Lower Stratosphere, Journal of Geophysical Research: Atmospheres, 128(4), e2022JD037699, https://doi.org/10.1029/2022JD037699, 2023.

Liu, N. and Liu, C.: Global distribution of deep convection reaching tropopause in 1 year GPM observations, Journal of Geophysical Research: Atmospheres, 121, 3824-3842, https://doi.org/10.1002/2015JD024430, 2016.

Liu, N., Liu, C. and Hayden, L.: Climatology and detection of overshooting convection from 4 years of GPM precipitation radar and passive microwave observations, Journal of Geophysical Research: Atmospheres, 125, e2019JD032003, https://doi.org/10.1029/2019JD032003, 2020.

(6) Lines 52-54: it is not clear what the authors mean here. What is the difference between convective overshooting and deep convection?

Answer: Thanks for your question, and Modifications have been made in the manuscript, shown as line 66-67. Rain top heights of more than 10 km are defined as deep convection, whose rain top heights are more than 14 km are defined as convective overshooting. Deep convection includes convective overshooting, but overall it's not as strong as convective overshooting.

(7) Lines 55-56: "of the polarimetric radar" should be "of polarimetric radar observations"

Answer: Thanks for your advice! Modifications have been made in the manuscript, shown as line 70.

(8) Line 62: revise "ways for detecting convective overshooting is to find pixels" to "way for detecting convective overshooting from satellite is to find pixels in infrared imagery"

Answer: Thanks for your advice! Modifications have been made in the manuscript, shown as line 76.

(9) Lines 66-68: Also, overshoots mix with relatively warm stratosphere air such that cold pixels are often diminish and not a reliable means to identify overshooting.

Answer: Thanks for your advice! We have added this to the manuscript, shown as line 83-84.

(10) Lines 83-84: because of what? This claim seems unsubstantiated to me. Synoptic evolution is typically slow and tropopause altitudes do not change rapidly (i.e., in periods <6 hr) in most circumstances. The varying latitude of the tropopause break, which is responsible for the band of high tropopause altitude deviation in Figure 1c, is a case where the tropopause could change rapidly, but it is also poorly constrained at such an abrupt transition.

Answer: Thanks for your reminder! We delete that sentence, shown as line 100-102.

(11) Line 88: "cold tropopause" should be "cold point tropopause". Also, as mentioned above, here and after "thermodynamic tropopause" should be "thermal tropopause".

Answer: Thanks for your advice! Modifications have been made in the manuscript, shown as line 106, 110, 117, 118.

(12) Lines 117-118: why choose June, July, and August only? Is it based on Liu et al. KuPR results?

Answer: Thanks for your question! On the one hand, observations and model simulations show that deep convection over land more frequently overshoot the tropopause during summer (June, July and August) and inject ice and water vapor into the lowermost stratosphere in midlatitude (Wang et al., 2023). On the other hand, due to limited space, only one season can be selected for in-depth research. In the future, we will specialize in the seasonal variation characteristics of convective overshooting.

Wang, X., Huang, Y., and Qu, Z. et al.: Convectively Transported Water Vapor Plumes in the Midlatitude Lower Stratosphere, Journal of Geophysical Research: Atmospheres, 128(4), e2022JD037699, https://doi.org/10.1029/2022JD037699, 2023.

(13) Section 2.2: what ERA5 products do you use. Specifically, what grid spacing (horizontal and vertical)? Those are important details to note regardless of how it is used.

Answer: Thanks for your advice! Modifications have been made in the manuscript, shown as line 145-146, 148-151.

(14) Lines 154-166: I don't find much value in this analysis.

Answer: Thanks for your advice! We rewrote this paragraph, shown as line 208-225.

(15) Line 159: "else region" should be "otherwise"

Answer: Thanks for your advice! Modifications have been made in the manuscript, shown as line 214.

(15) Line 195: "allow" should be ", which allows"

Answer: Thanks for your advice! Another referee also pointed out this issue. Combining your two suggestions, modifications have been made in the manuscript, shown as line 256.

(16) Lines 196 & 198: "penetrate *the* troposphere" should be "reach the stratosphere"

Answer: Thanks for your advice! Modifications have been made in the manuscript, shown as line 257.

(17) Lines 199-201: unnecessary - recommend deleting

Answer: Thanks for your advice! Modifications have been made in the manuscript, shown as line 261-263.

(18) Line 204: "with regionally different" should be "varying regionally (Table 1)"

Answer: Thanks for your advice! Modifications have been made in the manuscript, shown as line 266.

(19) Lines 204-206: no need to repeat numbers from the table here. Just describe the differences.

Answer: Thanks for your advice! Modifications have been made in the manuscript, shown as line 266-269.

(20) Section 3.2.2. The diagrams referred to here as DPDH would be more appropriately referred to the community standard of CFADs (contoured frequency by altitude diagrams). Also, there are many instances of "the zero level". What is meant

by this? Do you mean the altitude where the temperature is 0 ℃? If so, that is not evidenced by any of the analysis that you show!

Answer: Thanks for your advice! "DPDH" have been modified to "CFADs", shown as line 276, 279, 280, 286, 428, 430, 786-791. And we have added the explanation of "the zero level" in the manuscript, shown as line 289-290.

(21) Line 217: delete "obviously"

Answer: Thanks for your advice! Modifications have been made in the manuscript, shown as line 280-281.

(22) Line 219: "peak 47" should be "peak near 47"

Answer: Thanks for your advice! Modifications have been made in the manuscript, shown as line 282.

(23) Line 222: "feature are" should be "character is"

Answer: Thanks for your advice! Modifications have been made in the manuscript, shown as line 286.

(24) Line 227: rather than more ice crystals, this could alternatively imply they are larger.

Answer: Thanks for your advice! Modifications have been made in the manuscript, shown as line 292.

(25) Line 231: "very" should be "much"

Answer: Thanks for your advice! Modifications have been made in the manuscript, shown as line 296.

(26) Line 233: "precipitation" should be "production"

Answer: Thanks for your advice! Another referee also pointed out this issue. Combining your two suggestions, modifications have been made in the manuscript, shown as line 298.

(27) Lines 304-345: This should all be removed based on the comment provided above.

Answer: Thanks for your advice! We remove that, shown as line 370-411.

(28) Line 350: "a more accurate algorithm". Based on what evidence?

Answer: Thanks for your question! After thinking about the question, we have changed this sentence to "a reliable algorithm". Here's and explanation of why this algorithm is reliable. First of all, the algorithm design is strictly based on the principle of the definition of convective overshooting (Rain top height higher than tropopause height), which ensures the accuracy of the algorithm in principle.

From the perspective of the data input of the algorithm, tropopause height calculated from ERA5 and rain top height from GPM DPR are reliable. We have compared tropopause height calculated from ERA5 with sounding observation (IGRA), occultation detection (COSMIC) and reanalysis data (JRA55 and MERRA2) (Sun et al., 2021). Results show that tropopause calculated from ERA5 is reliable. Rain top height data here we use mainly relies on GPM KuPR's echo top height and KuPR is good at detecting intense precipitation like convective overshooting (Kojima et al., 2012), which guarantee the accuracy of the detection of rain top height. Based on the principle of the algorithm and the input data, the detecting method in this manuscript is reliable.

Sun, N., Fu, Y., Zhong, L., Zhao, C. and Li, R.: The Impact of Convective Overshooting on the Thermal Structure over the Tibetan Plateau in Summer Based on TRMM, COSMIC, Radiosonde, and Reanalysis Data, Journal of Climate, 34, 8047-8063, https://doi.org/10.1175/JCLI-D-20-0849.1, 2021.

Kojima, M., and Coauthors: Dual-frequency precipitation radar (DPR) development on the global precipitation measurement (GPM) core observatory, Earth Observing Missions and Sensors: Development, Implementation, and Characterization II, H. Shimoda et al., Eds., International Society for Optics and Photonics (SPIE Proceedings, Vol. 8528), 85281A, https://doi.org/10.1117/12.976823,2012.

(29) Line 356: delete "obviously"

Answer: Thanks for your advice! Modifications have been made in the manuscript, shown as line 422.

(30) Lines 359-360: "differences. And" should be "differences, and"

Answer: Thanks for your advice! Modifications have been made in the manuscript, shown as line 426.

(31) Line 364: "And the" should be "The" & "obviously" should be "obvious"

Answer: Thanks for your advice! Modifications have been made in the manuscript, shown as line 430.

(32) Lines 384-397: remove

Answer: Thanks for your advice! We remove that, shown as line 451-464.

---

## Author Response (AR2)

Dear referee,

Thank you again for reading the revised manuscript and our response to your comments so carefully! We are very impressed by your serious and responsible approach to the manuscript. We have read your suggestions carefully, and modifications have been made in the manuscript. Our replies are shown as follows.

1. Elaborate upon the need and perceived importance of improving understanding of precipitation physics in overshooting convection in the Introduction. Right now, the motivation is limited to improving models and understanding of the efficiency of vapor transport to the UTLS. Given the connections to severe and/or hazardous weather, including precipitation extremes, I believe the motivation can be much broader than it currently is.

Answer: Thanks for your advice! We've broadened the motivation, and added parts are shown as line 56-76. And following references have been added.

Reynolds, D. W.: Observations of damaging hailstorms from geosynchronous satellite digital data, Monthly Weather Review, 108, 337-348, https://doi.org/10.1175/1520-0493(1980)108<0337:OODHFG>2.0.CO;2, 1980.

McCann, D. W.: The enhanced-V: A satellite observable severe storm signature, Monthly Weather Review, 111, 887-894, https://doi.org/10.1175/1520-0493(1983)111<0887:TEVASO>2.0.CO;2, 1983.

Negri, A. J. and Adler, R. F.: Relation of satellite-based thunderstorm intensity to radar-estimated rainfall, Journal of Applied Meteorology and Climatology, 20, 288-300, https://doi.org/10.1175/1520-0450(1981)020<0288:ROSBTI>2.0.CO;2, 1981.

Fujita, T. T.: The Teton-Yellowstone tornado of 21 July 1987, Monthly Weather Review, 117, 1913-1940, https://doi.org/10.1175/1520-0493(1989)117<1913:TTYTOJ>2.0.CO;2, 1989.

Kellenbenz, D. J., Grafenauer, T. J. and Davies, J. M.: The North Dakota tornadic supercells of 18 July 2004: Issues concerning high LCL heights and evapotranspiration, Weather and forecasting, 22, 1200-1213, https://doi.org/10.1175/2007WAF2006109.1, 2007.

Brunner, J. C., Ackerman, S. A., Bachmeier, A. S. and Rabin, R. M.: A quantitative analysis of the enhanced-V feature in relation to severe weather, Weather and Forecasting, 22, 853-872, https://doi.org/10.1175/WAF1022.1, 2007.

Setvák, M., Lindsey, D. T. and Novák, P. et al.: Satellite-observed cold-ring-shaped features atop deep convective clouds, Atmospheric Research, 97, 80-96, https://doi.org/10.1016/j.atmosres.2010.03.009, 2010.

Dworak, R., Bedka, K., Brunner, J. and Feltz, W.: Comparison between GOES-12 overshooting-top detections, WSR-88D radar reflectivity, and severe storm reports, Weather and Forecasting, 27, 684-699, https://doi.org/10.1175/WAF-D-11-00070.1, 2012.

Bedka, K. M.: Overshooting cloud top detections using MSG SEVIRI Infrared brightness temperatures and their relationship to severe weather over Europe, Atmospheric Research, 99, 175-189, https://doi.org/10.1016/j.atmosres.2010.10.001, 2011.

Lane, T. P. and Sharman, R. D.: Gravity wave breaking, secondary wave generation, and mixing above deep convection in a three‑dimensional cloud model, Geophysical Research Letters, 33. https://doi.org/10.1029/2006GL027988, 2006.

Lane, T. P., Sharman, R. D., Clark, T. L. and Hsu, H. M.: An investigation of turbulence generation mechanisms above deep convection, Journal of the atmospheric sciences, 60, 1297-1321, https://doi.org/10.1175/1520-0469(2003)60<1297:AIOTGM>2.0.CO;2, 2003.

2. Increase emphasis on the goals of the current study in the Introduction. For example, while tropopause definition is important for overshooting identification, the departure to discuss this in the sixth paragraph of the Introduction increases focus on the overshooting process and stratospheric impacts (which I took to be the most important goal of the effort and led to my disappointment that detailed characteristics within the overshoots were not the focus of the analysis). This content should likely move to the methods to justify use of the chosen tropopause definition.

Answer: Thanks for your advice! Modifications have been made in the manuscript, shown as line 107-127, 143-144, 165-186.

3. Increase motivation for focusing primarily on East China. Since you are not limited to this region based on the datasets you employ, the decision to focus on it should be a bit better justified than in the present draft of the paper.

Answer: Thanks for your advice! Modifications have been made in the manuscript, shown as line 128-144.

4. For the analysis and associated discussion, the altitude of the freezing level should be indicated in your figures (perhaps by average and standard deviation for your samples) because none of the current analyses demonstrate where that is!

Answer: Thanks for your advice! The altitude of the freezing level have been indicated in figure 6, 7,10.